# Experiments and Finite Element Simulations of Composite Laminates Following Low Velocity On-Edge Impact Damage

**DOI:** 10.3390/polym14091744

**Published:** 2022-04-25

**Authors:** Wenjun Xu, Longquan Liu, Wu Xu

**Affiliations:** Laboratory of Aircraft Structure and Strength, Shanghai Jiao Tong University, Shanghai 20024, China; xuwenjun@sjtu.edu.cn (W.X.); xuwu@sjtu.edu.cn (W.X.)

**Keywords:** composite materials, on-edge impact, compression after edge impact, finite element simulation

## Abstract

Composites are widely used in aircraft structures that have free edges and are vulnerable to impact events during manufacturing and maintenance. On-edge impact may have a great contribution in terms of the compression strength loss of composites, but the influence remains unclear. This paper presents experiments and simulations of carbon-fiber-reinforced plastic (CFRP) materials with on-edge impact and compression after edge impact (CAEI). On-edge impact damage was introduced to the composite laminates through the drop weight method with 4, 6, 8 and 10 J impact energies, respectively. A special guide-rail-type fixture was used in the compression tests in which strain–force and load–displacement relationships were obtained. A continuous-step finite element model was proposed to simulate impact and compression. Continuum shell elements and Hashin failure criteria were used to simulate in-ply damage, and interlaminar damage was modelled by cohesive elements. The model was validated by correlating the experimental and numerical results. The investigation results revealed the relationships of the damage size and residual strength with the different impact energies. The crack length and delaminated area grow with the increase in impact energy. The residual compressive strength follows a downward trend with increasing impact energy.

## 1. Introduction

Due to the attractive mechanical characteristics of the high specified stiffness and strength of composites’ materials, they are widely used in the aviation, space, transportation and wind power industries. Unfortunately, composites’ materials have high impact damage susceptibilities. Low velocity impact (LVI) on composites’ materials could cause such damage that there is nearly invisible damage morphology on the exterior surface but extensive internal damage, named barely visible impact damage (BVID). The horrific damage caused by LVI could result in drastic reductions in the residual compressive strength of the composite materials; hence, the study of LVI is becoming more and more critical to composite materials and attracting more and more attention in recent years [1,2]. For example, Umair et al. [3] conducted low velocity drop weight impact testing on natural-fiber-reinforced polymeric composites and investigated the maximum force, displacement and impact energy absorption during the impact events. The study revealed the effect of the weave structure and quantity of glass microspheres on the LVI properties. Hussain et al. [4] determined the better combination of hybridization and matrix against the LVI performance of hybrid-reinforced fiber metal laminates by conducting a drop weight impact test and compared the response of different factors on the LVI performance. Zangana et al. [5] investigated the LVI behavior of a novel corrugated core sandwich composite by the experimental and numerical method. The impact behavior, energy absorption ability and failure mode of a trapezoidal composite were studied to determine its vulnerability under low velocity impact. A finite element modelling method was proposed to predict the impact capacity of a trapezoidal composite corrugated core sandwich. The low velocity impact behavior of a novel fiber composite sandwich was also investigated [6]. To achieve the superior performance in specific stiffness/strength, energy absorption and core damage of trapezoidal corrugated core sandwich composite, the combination of different fibers was studied by the low velocity impact method. This study revealed that hybridized sandwiches with high-performance fiber perform well when subjected to impact and proposed an empirical relationship to predict the residual strength.

Previous studies are typically restricted to the on-face impact and its effects on composite structures. However, impact events are often on-edge in practice. For example, among the airplane components, such as the center wing box, there are many free edge stiffeners exposed to a possible LVI circumstance, which include tool drop, accidental or runway debris impact during manufacturing and in-service processes. The schematic diagram of on-edge impact is shown in Figure 1. This kind of on-edge impact damage may lead to more severe damage within the laminates than on-face impact, thus causing the catastrophic loss of the residual compressive strength of composites [7]. In order to preferably determine the maintenance threshold value and guide the design of composite structures, the damage resistance and damage tolerance performances need to be determined by the investigation of on-edge impact.

On-edge impact has attracted much attention. Malhotra et al. [8] performed their study including both experiments and numerical simulations of near-edge and on-edge impact damage of the laminates. Their study explored the extent and mechanisms of the edge impact of the quasi-isotropic composite laminates, and they discovered that the damage width arises with the increase in incident energy impact and a single delamination appears with the near-edge impact but multiple, longer delaminations appear with the on-edge impact. The vulnerability of composite laminates to on-edge impact was demonstrated by their study results.

Some other studies paid attention to the compression after the impact (CAI) of composite structures. For example, Rhead et al. [9] reported a semi-analytical model based on fracture mechanics to predict the residual compressive strength of laminated composite material after on-edge impact. They validated their theoretical results by testing three impacted coupons’ compression experiments and comparing with the analytical results. The maximum difference of the initial propagation strain of damage between the theoretical values and experimental values is less than 10%. Thorsson et al. [10,11] studied the influences of different impact angles and energies on a specific polymer matrix composite laminate subjected to edge impact and CAI. Non-destructive and destructive post-impact inspections were applied to measure the damage after impact. Using an improved version of the combined loading compressive method to capture residual compressive strength is demonstrated to be more effective than the industry standard for compression after impact. A finite element model was introduced for simulating the dynamic response and residual compressive strength after on-edge impact. Besides composite laminates, the damage tolerance of wing-relevant stiffened composites’ panels was investigated by Li et al. [12,13,14]. The stiffened composite plates with two kinds of stiffeners, which were T-shaped and I-shaped, were subjected to low velocity impact and compression after impact to investigate their CAI strength and failure modes. Different impact damages on purposefully selected locations, such as the outer surfaces of the panels and free edges of stiffeners, were introduced to specimens. The conclusion that the damage to the stiffener edges caused by impact is more severe than to the panel was drawn from the study. The experimental results revealed the triggers of the final failure of a stiffened composite plate, including local buckling, following the damage growth and final fracture of the stiffener due to edge impact, which also determines the ultimate carrying load. Two phenomenologically based mechanical finite element models were also proposed to predict the residual compressive strength and failure modes. In order to reduce the research complexity of stiffened composite structures, Ostré et al. [15,16,17] conducted a series of experimental investigations and numerical simulations on CFRP. They used an original edge impact setup, which clamps the half of the laminate specimen to maintain another half-height free edge to impact. The impact results show that the fiber properties control the initial impact stiffness and a specific “crushing plateau”, whose value is approximately equal to an average stress multiplied by an average projected area of impact. From the CAEI tests, regarding the mechanisms, it can be observed that the propagation of a compressive fiber failure is a critical factor regarding the laminate residual strength after edge impact. A discrete-ply-model-based finite element model was suggested to simulate edge impact and CAEI. This model shows great correlation with the experiment on the following phenomena: permanent indentation, force-displacement curve during impact, crack length, delamination, stress–displacement curve from CAEI, stress-out-of-plane displacement curve and CAEI final failure. Some other studies related to low velocity on-edge impact and compression after edge impact were presented recently. Arteiro et al. [18] presented a composite damage model based on ply-by-ply discretization to simulate the LVI and CAEI behavior of laminated composites. Liu et al. [19] studied near-edge/on-edge LVI and CAEI problems with comprehensive procedures. By using theoretical analysis, numerical simulation and experimental investigation, the damage size, residual strength and damage mechanism of the composite laminates were revealed. The influence of the thickness, layering sequence and impact energy of T300/69 laminates was described.

In previous research, there are a few studies regarding the on-edge impact of composite structures, but the repeatability of the impact location is low since the laminates are very thin, and, thus, the consistency of the test results was poor. This paper investigates the relationships of impact energy, crack length after impact and residual compressive strength after on-edge impact with both experimental and numerical methods. A unique on-edge impact tool is used in the impact tests, and a guide-rail-type CAEI tool is used in the compression tests to obtain the residual compressive strength. The relationships between the impact energies with the damage size and residual strength of the composite laminates are studied. In the finite element model, the Hashin failure criterion on the continuum shell element is used to determine the intralaminar damage, and the zero-thickness cohesive element is used to characterize the interlaminar crack initiation and propagation. A good correlation of the experimental results and numerical simulation results is presented. The outcomes of this research can help in improving the designs of composite structures that are vulnerable to on-edge impacts.

## 2. Experimental Method

### 2.1. Specimen

Carbon-fiber-reinforced plastic composites X850/IM+ with quasi-isotropic lay-up ([45/0/−45/90]_3s_) were chosen to conduct the impact and CAEI tests. The ply thickness is 0.185 mm. The specimens are all of a length of 200 mm and a width of 75 mm. The material properties of the lamina are listed in Table 1, where E_1_ is longitudinal modulus, E_2_ is transverse modulus, ν_12_ is Poisson’s ratio, G_ij_ is shear modulus corresponding to different directions, X_T_ and X_C_ are longitudinal tensile and compressive strength, respectively, Y_T_ and Y_C_ are transverse tensile and compressive strength, respectively, S_XY_ and S_YZ_ are longitudinal and transverse shear strength, respectively.

The properties of cohesive layers are listed in Table 2, where η is estimated value. The meanings of symbols in Table 2 are that K_n_, K_s_ are interface stiffness, σ_n_ is nominal strength, σ_s_ is transverse strength, G_ⅠC_, G_ⅡC_ and G_ⅢC_ are mode Ⅰ, Ⅱ and Ⅲ fracture energy, respectively.

### 2.2. On-Edge Impact Test

On-edge impact tests were carried out using a CEAST 9350 drop weight impact test machine referring to the ASTM D7136 standard [20]. Four different impact energy levels, which are 4 J, 6 J, 8 J and 10 J, respectively, and three repeated tests were conducted with each impact level, which are shown in Table 3.

A special test setup was established as shown in Figure 2. There are two impactors. A steel flatten impactor is on the top and connected with the testing machine. A steel hemispherical tup is at the bottom and contacted with the edge of the specimen. The hemispherical tup consists of a 16 mm diameter steel rod with a hemispherical end of the same diameter and a hardness of 62 HRC. The mass of the flatten impactor and the hemispherical tup are 5.565 kg and 0.100 kg, respectively. Both the impactor and the tup can only move along the vertical direction (Y direction in the Figure 2).

The testing specimens were fixed along the bottom edge and the two ends, which are shown in Figure 2. The impact location was guaranteed to be at the right location (in the center of the side surface of the specimens both along the length direction and thickness direction).

The site photo the test setup is shown in Figure 3, which shows the positional relations of the specimen, impactors and testing jig. During impacting, the flatten impactor’s displacement, force and velocity were measured in real time with the built-in sensors in the testing machine. The data were recorded at a sampling rate of 1000 data per second.

### 2.3. CAEI Test

Quasi-statically compressive tests were performed referring to the ASTM D7137 standard [21] to obtain the CAEI performances of the composite laminates. A computer-controlled material testing machine was used to compress the specimens with impact damage, and a set of testing jig was used to support the specimens and prevent the specimens from buckling during compression. The CAEI fixture has been designed as a guide-rail-type tool in order not to disturb the damage due to the on-edge impact. The residual compressive strength can be sensitive to the on-edge impact by using this fixture. The top and bottom clips (x-direction) of this fixture could constrain the degree of freedom of the specimen, except loading direction. Two guiding rails also make a contribution. The two guiding rails are composed of two knives, which let the specimen move in the compressive loading direction only. The schematic of the compression fixture is shown in Figure 4.

Four strain gauges were pasted on both sides of specimens to monitor the strain values during compression and to center-align the specimen with the loading machine. The compressive load and displacement were measured and recoded by the transducers embedded in the testing machine.

The specimens were compressively loaded in displacement control mode with a constant speed of 1.0 mm/min, which kept proceeding until the ultimate load decreased by 40%. The environmental temperature was maintained at (23 ± 5) °C, and the relative humidity was maintained at (55 ± 5)% during tests. The sampling rates of the load, displacement and strains were all set to be 10 Hz.

## 3. Numerical Simulation

### 3.1. Finite Element Model

The finite element model of the composite laminates under on-edge impact and compression after on-edge impact was established using the commercial code, Abaqus, and the Explicit algorithm was used to simulate the impact and compression process. The failure of the composite laminate was composed of the intra-lamina damage and inter-lamina damage. The former was simulated using the Hashin failure criterion and the latter was simulated using the cohesive zone method. Each lamina of the specimen was modeled by an eight-node in-plane reduced integral continuum shell element, which is the SC8R element in Abaqus. The delamination was simulated with zero-thickness cohesive elements between two consecutive plies. For the accurateness of the finite element simulation, fine mesh was used to model the impact area, which is represented by a semicircle with the radius of 30 mm. The size of the elements in the semicircle area is 0.5 mm in length, and there are 32 elements on the circumference. Elements 1.2 mm in size were arranged away from the semicircle area, and the size of these elements increases with the distance from the impact point by using a nodal bias. The geology and mesh size of the finite element model are shown in detail in Figure 5.

### 3.2. Intralaminar Failure Criterion

The damage model of composite was established by simulating the in-layer failure and interlayer delamination. The stress of the lamina model was calculated by classical composite laminate plate theory. Losses in strength and stiffness of composite laminates were considered as failures. The failure modes of fiber damage fracture, matrix damage fracture and delamination failure could be drawn from the simulation. According to the Hashin failure criterion [22], the damage of the fiber and matrix in a composite lamina can be identified, respectively, as follows:Damage of fiber

Fiber tensile (σ_xx_ ≥ 0)
(1)$${\left(\frac{\sigma_{\mathrm{xx}}}{X_{T}}\right)}^{2}+{\left(\frac{\sigma_{xy}}{S_{12}}\right)}^{2}+{\left(\frac{\sigma_{\mathrm{xy}}}{S_{12}}\right)}^{2}\geq 1$$

Fiber compression (σ_xx_ < 0)
(2)$${\left(\frac{\left|\sigma_{\mathrm{xx}}\right|}{X_{C}}\right)}^{2}\geq 1$$

2.Matrix failure

Matrix cracking (σ_xx_ + σ_zz_ ≥ 0)
(3)$${\left(\frac{\sigma_{\mathrm{yy}}+\sigma_{\mathrm{zz}}}{Y_{T}}\right)}^{2}+{\left(\frac{\sigma_{\mathrm{xy}}}{S_{12}}\right)}^{2}+{\left(\frac{\sigma_{\mathrm{xz}}}{S_{13}}\right)}^{2}+\left(\frac{\sigma_{\mathrm{yz}}^{2}-\sigma_{\mathrm{yy}}\sigma_{\mathrm{zz}}}{S_{23}^{2}}\right)\geq 1$$

Matrix extrusion (σ_yy_ + σ_zz_ < 0)
(4)$$\frac{1}{4}{\left(\frac{\sigma_{\mathrm{yy}}+\sigma_{\mathrm{zz}}}{S_{12}}\right)}^{2}+{\left(\frac{\sigma_{\mathrm{xy}}}{S_{12}}\right)}^{2}+{\left(\frac{\sigma_{\mathrm{xz}}}{S_{13}}\right)}^{2}+\left(\frac{\sigma_{\mathrm{yz}}^{2}-\sigma_{\mathrm{yy}}\sigma_{\mathrm{zz}}}{S_{23}^{2}}\right)\;\;+\left(\frac{\sigma_{\mathrm{yy}}+\sigma_{\mathrm{zz}}}{Y_{C}}\right)\left[\frac{1}{4}{\left(\frac{Y_{C}}{S_{12}}\right)}^{2}-1\right]\geq 1$$

σ_ij_ in the formula represents the stress component of each element in the material principle coordinates.

### 3.3. Interlaminar Damage

Zero-thickness cohesive element was used to simulate delamination damage of the composite laminates [23]. The interface of adjacent plies would be fully degraded when the energy dissipated equals to the fracture toughness. The propagation criterion was implemented under mixed-mode loading conditions, which is the B-K criterion. The value of η, which is power exponent, is 1.45.
(5)$$G_{C}=G_{IC}+\left(G_{\mathrm{II}C}-G_{IC}\right){\left(\frac{G_{\mathrm{II}}}{G_{I}+G_{\mathrm{II}}}\right)}^{\eta }$$

### 3.4. Impact Simulation

The dynamic analysis in the Abaqus Explicit module was used in the simulation of the on-edge impact process. The details of the boundary conditions of the finite element model are shown in Figure 6a. The bottom edge and two longitudinal ends (x-direction) of the laminates were fully constrained due to the characteristics of the impact fixture. The impactor of the drop tower was represented by rigid shell elements. Its shape is a 40 mm diameter cylinder with a point mass of 5.565 kg. The velocity values calculated by the incident energy were assigned to the impactor. The tup of the impact fixture was modeled as a rigid body and assigned the mass of 0.100 kg. The tup was hemisphere-moldered and cylindrical, with a diameter of 16 mm. The surface-to-surface contact was used to simulate the interactive relationships between the impactor with the tup and the tup with the edge of the specimens. A friction coefficient of 0.3 was used for the contacts. The impactor and tup were constrained to move in the vertical direction (y-direction) only. The time of this step is 10 milliseconds to make the tup rebound and release the oscillations.

### 3.5. Compression Simulation

The CAEI simulation was also simulated using the Abaqus Explicit method to avoid convergence problems. The details of the boundary conditions of the compressive step are shown in Figure 6b. The primary boundary conditions used in the impact analysis were moved and new boundary conditions were introduced in order to add a compressive load. The right edge of the laminate model was completely constrained. The bottom edge and central line of the laminate model along the longitudinal direction (x-direction) were constrained to move in the x direction. A reference point was built to tie with the nodes of the left edge of the laminate model. A displacement was assigned to the reference point until the laminate failure. This step lasts 150 milliseconds to be representative of the static compression.

## 4. Results Analysis and Discussion

### 4.1. Impact Damage

It is well known that LVI can cause BVID in composite laminates. There are many ways to characterize the severity of the damage. Based on the findings of previous research, the role of delamination is critical to damage tolerance after impact, and the compressive strength can be dramatically reduced [24,25]. In this study, the severity of impact damage is characterized by the crack length and delaminated area, which are schematically shown in Figure 7.

The impact damage on the edge of the specimens is shown in Figure 8. Indentations and cracks could barely be seen by naked eyes on all the impacted specimens, and the values of them are under the BVID thresholds [26].

The impact damage sizes were established through the non-destructive inspection (NDI) method, and the A GE Phasor XS phased array ultrasound inspection device was used to detect the damage size. The inspector uses high frequency sound waves as a probing medium to detect cracks. The sound waves travel through the specimen with attendant energy loss and are reflected at material-crack interfaces. The delamination shapes of the laminates are nearly semi-ellipse, and the long axis is vertical to the impact direction. Table 4 lists the impact damage sizes of the different impact cases.

Figure 9 shows the relationships of the crack length and delamination area with the impact energies. From Figure 9, it can be seen that both of the two damage features increase with the increase in the impact energies.

### 4.2. CAEI Experimental Results

The post-impact specimens were submitted to CAEI to obtain their residual compressive strength. The strain, ε_xx_, was measured to guarantee the correct progression of the compression test. The location arrangement of the strain gauges is shown in Figure 4. The strain–load curves analysis indicates that the compression load was applied stably and coaxially to the CAEI specimens. The strain–load curves of the CAEI tests are shown in Figure 10. The maximum compressive strain (x-direction) could reach around 6000 μ. There is furcation of the strain that occurred at some time points where the outer sublayers of the laminates buckled, and this was accompanied by noise.

The compressive failure results are shown in Figure 11. With increasing compressive load, the minor sub-laminates in the impact area buckled and a specific broken noise was heard simultaneously. Impact damage is the eventual failure of the impacted laminate not due to the global buckling.

From a quantitative point of view, the load displacement curves show that the compression force value increases gradually, then the force value reaches a peak and suddenly drops drastically. Besides, the stiffness is similar for the 0, 4, 6, 8 and 10 J impact energy levels, with 0 J meaning that the values are unimpacted. The representative load–displacement curves of the specimens are shown in Figure 12. The ultimate compressive load at the highest impact energy, 10 J, decreases by almost 31.4 percent compared to the undamaged ultimate compressive load. Additionally, the ultimate compressive load at the lowest impact energy, 4 J, decreases by almost 16.5 percent compared to the undamaged ultimate compressive load. It can be seen that the ultimate compressive load decreases with an increase in the incident energy, and there is a negative correlation between residual strength and crack length, as shown in Figure 13. It is roughly said that 1 J/mm of incident energy could reduce the residual compressive strength by 15.5%. It is worth noting that the relationship of residual compressive strength and crack length is similar to that of residual compressive strength and delaminated area.

### 4.3. Finite Element Simulation Results

The main goal of the finite element simulation is to catch the relatively accurate damage size of the specimen after the on-edge impact and the residual compressive strength of the CAEI.

(1)Damage morphology of simulation results

When the incident energy is 4 J, the impact damage morphology of the finite element simulation is shown in Figure 14. From Figure 14, it can be seen that the primary damage modes include permanent indentation, cracks and delamination. The fiber failure, matrix crack and delamination are responsible for the impact damage. From the results of the finite element simulation, delamination is the most serious impact damage.

The failure of the CAEI is shown in Figure 15. After being on-edge-impacted, there is a large area of delamination. It can result in a decisive influence on the residual compression strength of the specimen. In the CAEI simulative process, delamination damage and matrix failure are the main forms of damage and are responsible for the failure of the composite laminates. The propagation direction of the damage was perpendicular to the direction of compressive loading when the compression load is applied. The progressive expansion of the introduced impact damage leads to the eventual failure of the laminates. Some sublayers located on the outer layer of the laminate buckled in the compression process.

(2)Damage size after impact

Table 5 shows the experimental and numerical results of the edge impact damage size and the difference between them. The crack lengths of the numerical simulation values are in good agreement with the test values, which is illustrated by the data in Table 5. All the absolute errors of the crack length values are within 10 percent.

(3)Residual compressive strength

The comparison of the residual compressive strength between the finite element simulation results and experimental results is displayed in Table 6. From Table 6, it can be concluded that the residual compressive strengths of the numerical simulation results are in good agreement with the test results. The absolute errors of the test results with the numerical results are less than 10%. Therefore, it can be considered that this kind of finite element simulative method of the LVI and CAEI is suitable and effective for predicting the CAEI strength.

(4)Longitudinal strain

The comparison of the finite element simulation values and test values of longitudinal strain is shown in Table 7. The strain values were obtained from the average experimental values of gauge1 and gauge3. The true strains from the finite element analysis results match well with the experimental values. The absolute error of the experimental value with the numerical simulative value is less than 5%, except that the maximum error is 9.53%.

## 5. Conclusions

In this paper, one of the main purposes was to be able to acquire the damage tolerance of laminated composite material. The effects of the impact energy on laminated composites were studied, and the residual compressive strength of laminated composites due to on-edge impact was investigated experimentally and numerically. To ensure that the impact location of the on-edge impact was right in the middle of the specimens‘ edge, a special test setup was proposed. The on-edge impact position of the specimens remained consistent. Four strain gauges were affixed to both sides of the specimens symmetrically to monitor and ensure the alignment of the specimens with the loading machine when the specimens were in compression load. From the strain–load curves, the specimens maintained good alignment during the compressive process. The sublayers at the outer layer of the specimens would buckle before the ultimate failure, and a broken noise was heard at the same time. This phenomenon means that a local buckling at the impact area occurred before the compressive failure. Overall, the following conclusions can be drawn from this paper:A novel impact fixture was designed to execute on-edge impact to make sure the location of the impact was perfectly on the middle edge of the specimen. The impact morphologies show that a good effect appears from the impact fixture. The impact damage extent was established through Ultrasonic C-scanning inspection of the laminates. Indentations and cracks could barely be seen by naked eyes on all the impacted specimens. There were multiple cracks after impact, and the longer cracks appeared in the outer layer of the laminates. The damage morphologies of the delamination of the laminates are nearly semi-ellipse, and the long axis is vertical to the impact loading direction. The crack length and delaminated area increase with the increase in impact energy.In the CAEI tests, the residual compressive strength is sensitive to the on-edge impact damage. Roughly 1 J/mm of incident energy could reduce the residual compressive strength by 15.5%. The load–displacement curves of the damaged laminates under various levels of incident energy reveal that the stiffness values of the different damage laminates were similar to each other. There was little effect of on-edge impact on stiffness. The compression force value of each curve would fall sharply when a curve reached its peak. The complete loss of the residual strength of damaged laminates will produce this phenomenon. From the CAEI damage scenario of the laminates, the sub-laminates located on the outer layer in the impact area buckled and a specific broken noise was heard simultaneously, and the sub-laminates in the central portion of the laminates were squashed at the end of the time. The ultimate compressive load follows a downward trend with increasing impact energy.A finite element simulation method was introduced for predicting the on-edge impact and CAEI. The on-edge impact damage and post-impact compression failure were drawn from the whole process of the numerical model. The intra-laminate failure was modeled with a continuum shell element and judged by the Hashin failure criterion. The inter-laminate interface was modeled by the zero-thickness cohesive element. The impactor of the impact device was modeled using a rigid body. In a general sense, the numerical simulation results are in good agreement with the experimental results.

## Figures and Tables

**Figure 1 polymers-14-01744-f001:**
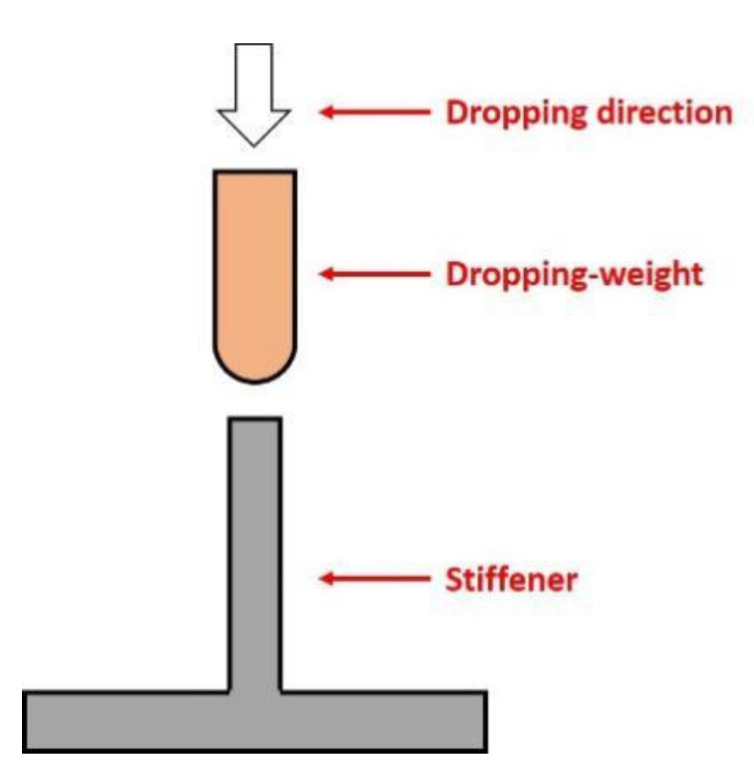
On-edge impact principle.

**Figure 2 polymers-14-01744-f002:**
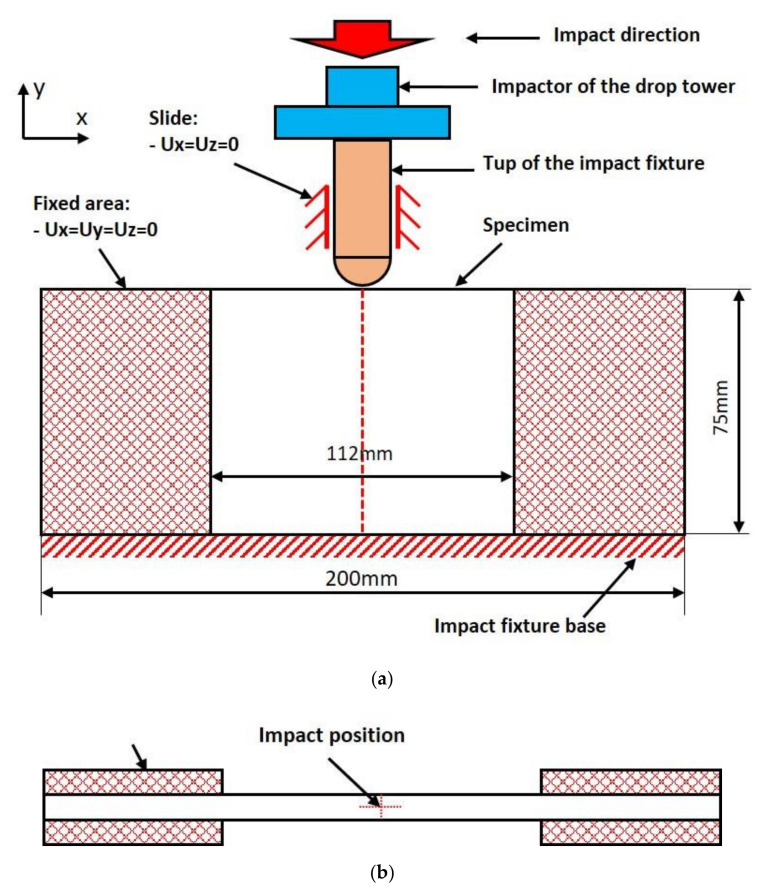
The setup of the impact tests. (**a**) Font view; (**b**) top view.

**Figure 3 polymers-14-01744-f003:**
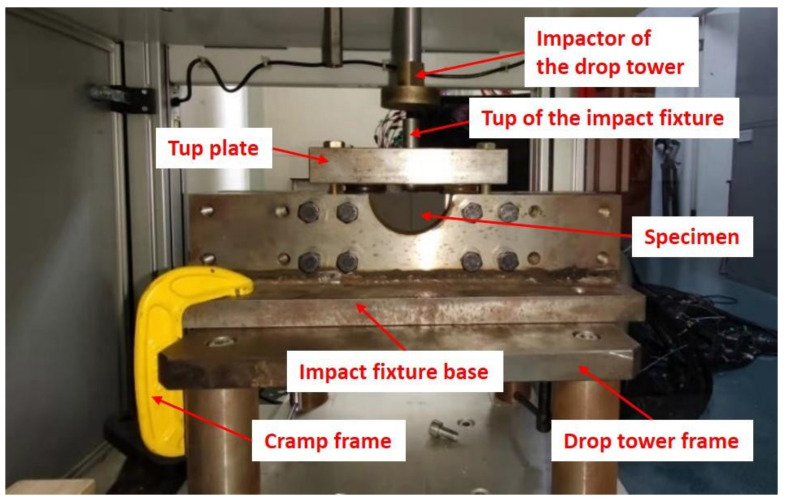
The site-photo of impact tests.

**Figure 4 polymers-14-01744-f004:**
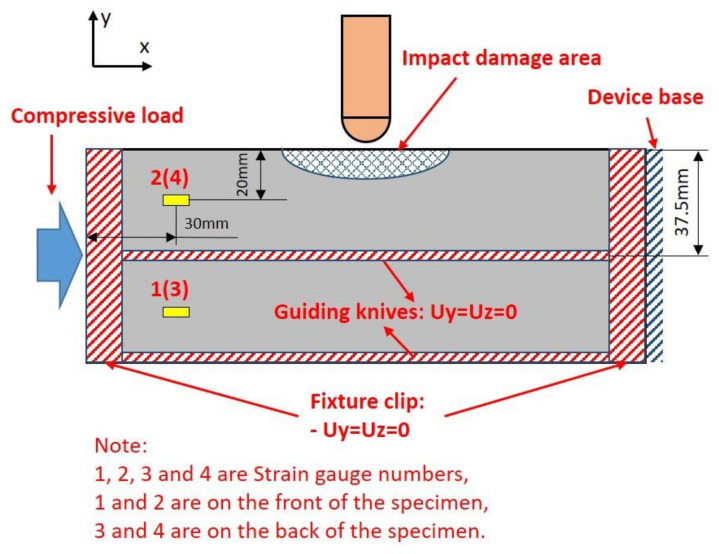
The setup of the compression tests.

**Figure 5 polymers-14-01744-f005:**
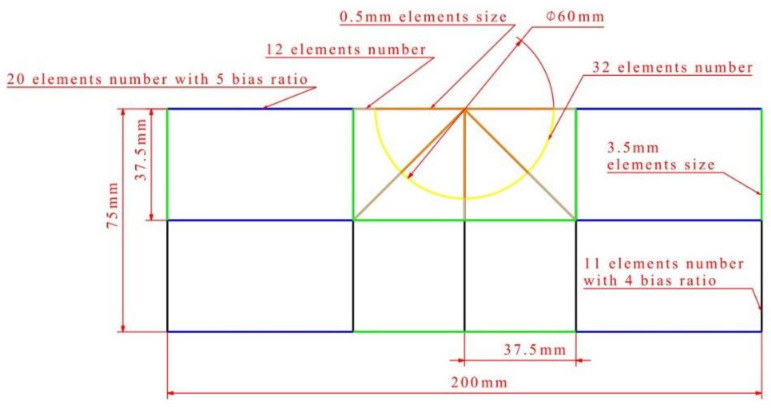
The geology and mesh size of the finite element model.

**Figure 6 polymers-14-01744-f006:**
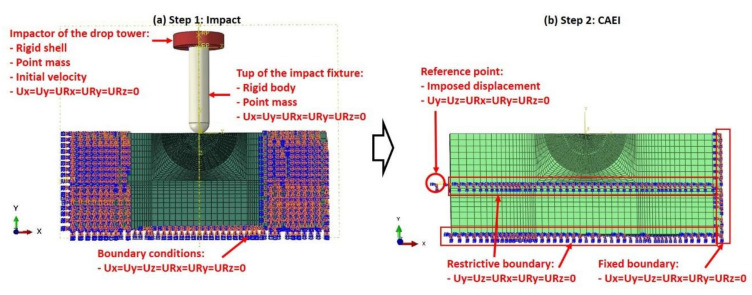
Finite element model strategy.

**Figure 7 polymers-14-01744-f007:**
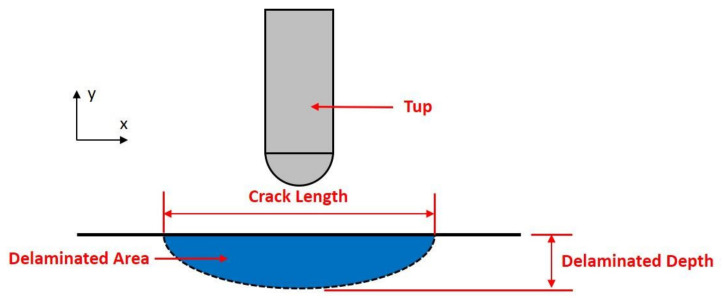
The schematic of the delamination morphologies.

**Figure 8 polymers-14-01744-f008:**
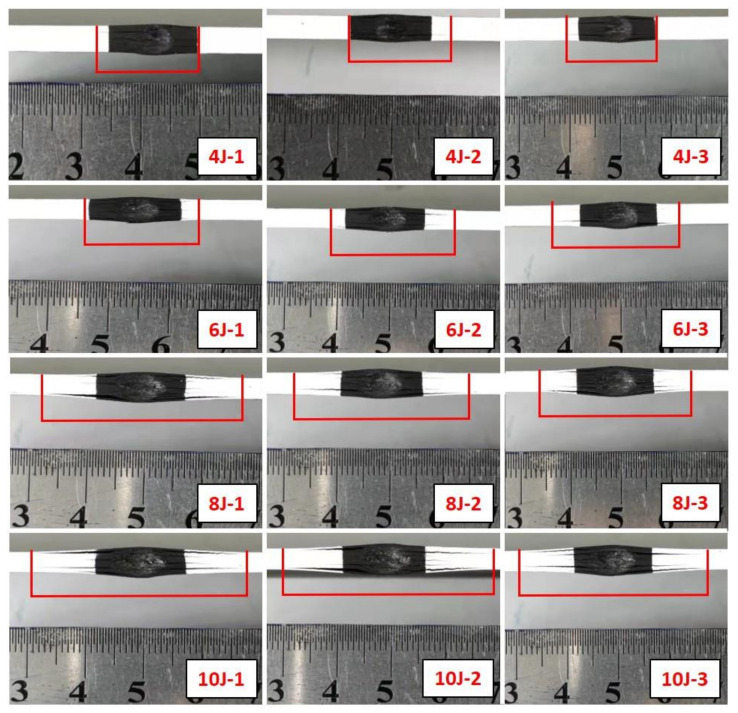
Surface damage of laminates.

**Figure 9 polymers-14-01744-f009:**
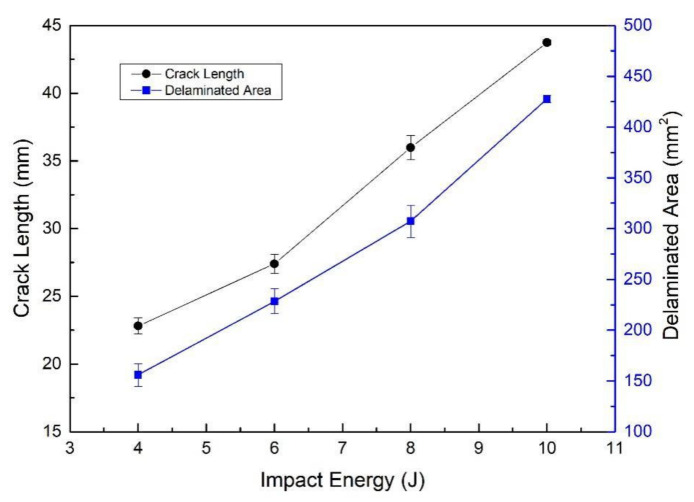
Impact delamination measurements.

**Figure 10 polymers-14-01744-f010:**
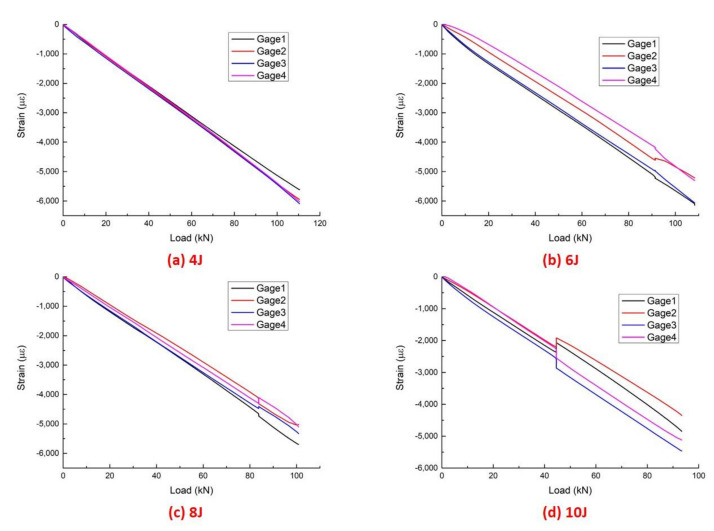
The strain–load curves of CAEI tests under different incident energy.

**Figure 11 polymers-14-01744-f011:**
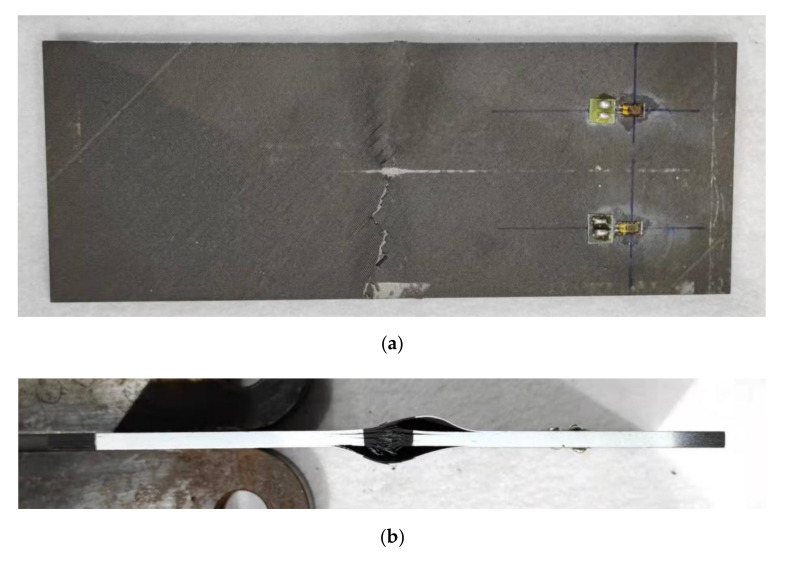
CAEI failure picture of the specimen at 10 J. (**a**) Front view; (**b**) impact edge side view; (**c**) non-impact edge side view.

**Figure 12 polymers-14-01744-f012:**
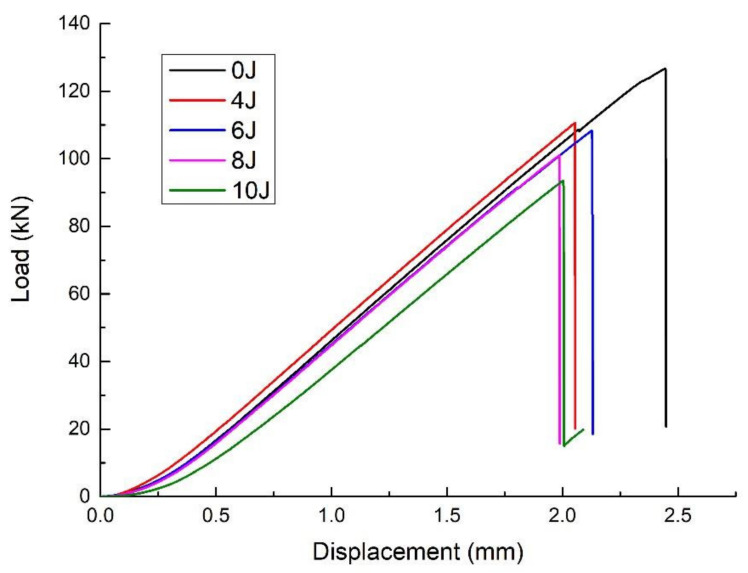
Load versus displacement curves of CAEI.

**Figure 13 polymers-14-01744-f013:**
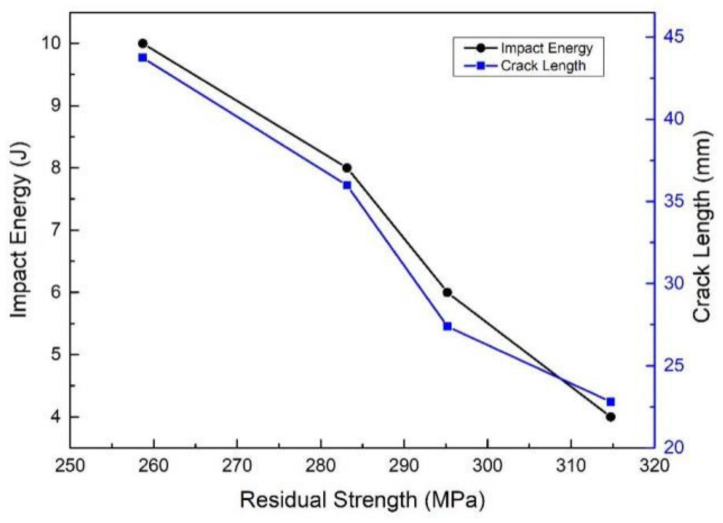
Residual strength versus impact level and crack length.

**Figure 14 polymers-14-01744-f014:**
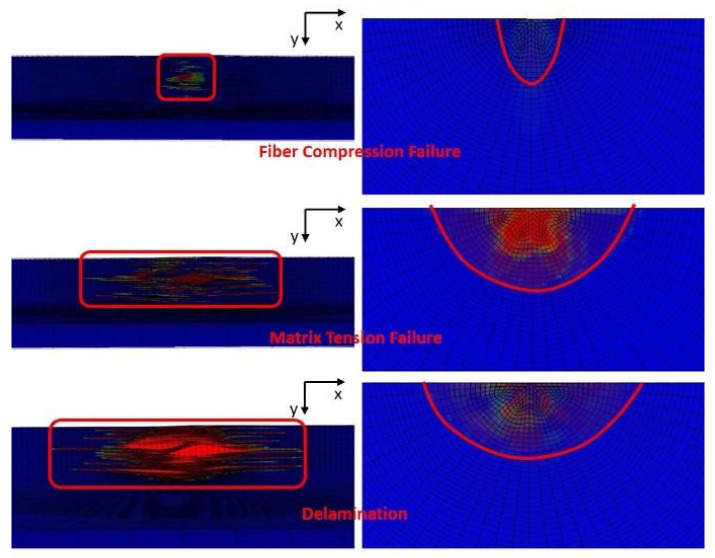
Impact damage of finite element simulation results.

**Figure 15 polymers-14-01744-f015:**
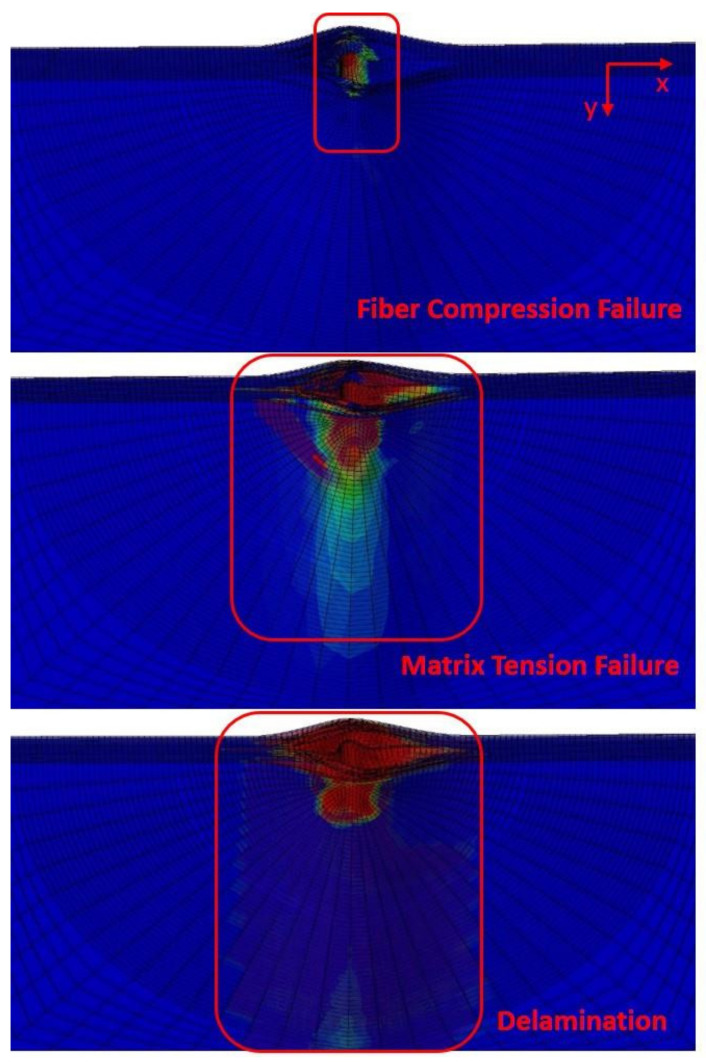
Damage scenario of CAEI of predicted results.

**Table 1 polymers-14-01744-t001:** Material properties of X850/IM + lamina.

E_1_ (GPa)	E_2_ (GPa)	ν_12_	G_12_ (GPa)	G_13_ (GPa)	G_23_ (GPa)
164	9.8	0.33	4.4	4.4	3.1
X_T_ (MPa)	X_C_ (MPa)	Y_T_ (MPa)	Y_C_ (MPa)	S_XY_ (MPa)	S_YZ_ (MPa)
3086	1492	73	275	96	105

**Table 2 polymers-14-01744-t002:** Strength properties of adhesive interface.

K_n_ (kN/m^3^)	K_s_ (kN/m^3^)	σ_n_ (MPa)	σ_s_ (MPa)	G_ⅠC_ (J/m^2^)	G_ⅡC_ (J/m^2^)	G_ⅢC_ (J/m^2^)
2.7 × 10^6^	1.1 × 10^6^	62	82	1.1	1.3	1.3

**Table 3 polymers-14-01744-t003:** Test plan.

Specimen Number *i* = 1,2,3	Tup Diameter (mm)	Impact Energy (J)
4 J-*i*	16	4
6 J-*i*	16	6
8 J-*i*	16	8
10 J-*i*	16	10

**Table 4 polymers-14-01744-t004:** Damage size of specimens.

Impact Energy (J)	Specimen Serial	Crack Lengths (mm)	Mean Crack Lengths (mm)	Delaminated Areas (mm^2^)	Mean Delaminated Areas (mm^2^)
4	4 J-1	24.69	22.81	177.73	156.02
	4 J-2	22.23		135.42	
	4 J-3	21.52		154.91	
6	6 J-1	24.69	27.40	191.49	228.45
	6 J-2	28.22		234.38	
	6 J-3	29.28		259.50	
8	8 J-1	38.10	35.98	316.43	307.29
	8 J-2	38.45		361.90	
	8 J-3	31.4		243.53	
10	10 J-1	42.69	43.74	425.60	427.68
	10 J-2	43.74		448.08	
	10 J-3	44.8		409.36	

**Table 5 polymers-14-01744-t005:** Comparison of the damage size.

Impact Energy (J)	Crack Length			Delaminated Area		
	Test Results (mm)	Numerical Results (mm)	Error (%)	Test Results (mm^2^)	Numerical Results (mm^2^)	Error (%)
4	22.81	21.50	−5.7	156.02	197.57	26.6
6	27.40	28.50	4.0	228.45	295.47	29.3
8	35.98	33.00	−8.3	307.29	396.55	29.0
10	43.74	39.50	−9.7	427.68	496.37	16.1

**Table 6 polymers-14-01744-t006:** Comparison of CAEI strength.

Impact Energy (J)	Test Values (MPa)	Numerical Simulative Values (MPa)	Error (%)
4	314.75	337.88	7.35
6	295.19	308.29	4.44
8	283.16	299.14	5.64
10	258.70	276.15	6.75

**Table 7 polymers-14-01744-t007:** Comparison of the residual strain.

Impact Energy (J)	Experimental Results (μ)	Numerical Results (μ)	Error (%)
4	5846	6403	9.53
6	6102	5817	−4.67
8	5510	5656	−2.65
10	5153	5235	1.59

## Data Availability

The data presented in this study are available on request from the corresponding author.

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
