# Peer review of "Experiments and Finite Element Simulations of Composite Laminates Following Low Velocity On-Edge Impact Damage"

_polymers, 2022, doi:10.3390/polym14091744_

Round 1

Reviewer 1 Report

  1. Introduction is not comprehensive; add more articles on low velocity impact of composites some examples are given below:                                                    a) Umair, M. Hussain, Z. Abbas, Effect of weave architecture and glass microspheres percentage on the low velocity impact response of hemp / green epoxy composites, J. Compos. Mater. (2021). https://doi.org/10.1177/0021998320987605.                                                          b) Hussain, A. Imad, Y. Nawab, A. Saouab, C. Herbelot, T. Kanit, Effect of matrix and hybrid reinforcement on fibre metal laminates under low – velocity impact loading, Compos. Struct. 288 (2022) 115371. https://doi.org/10.1016/j.compstruct.2022.115371.
  2. Please recheck the spelling mistakes throughout the manuscript, specifically line numbers 150, 351, and 353 the word setup and gauge have typing errors respectively.
  3. Why stiffness is similar for 0, 4,6,8, and 10 joules impact force, please justify it.
  4. Please explain Figure 3 with proper tagging.
  5. Please explain the crack detection mechanism through an Ultrasonic detector.
  6. In Figure 8 strain-load curves of CAEI tests for 4J and 8J are almost the same but at 6J and 10 J the strain variation between gauges is different, please justify this variation in results.
  7. Figure 11 is not clear, please properly label it.

Reviewer 2 Report

Comments

This paper investigate the composite laminates following low velocity on-edge impact damage. The outcome of the paper is interesting however, there are several aspects that need to be improved. The reviewer can onlyrecommend for publication if the author satisfactorily address the following major comments in the revised version.

  1. In Fig. 14, how the experimental failure mode was validated by FE simulation?
  2. The research gap from the literature review should be clearly presented.
  3. The research questions and justification of selecting variables should be highlighted.
  4. Which test standards was considered in this study? How many replicate samples were tested in each category?
  5. The failure mechanism of the specimen should be discussed more clearly.
  6. The novelty of the study should be highlighted more clearly at the end of introduction section. How this study is different from the published study in literature?
  7. How the outcome of this study will benefit researchers and end users? This need to be highlighted in introduction or end of conclusion.
  8. The low velocity impact of composites is interesting but not novel. Therefore, the recent application in this area should be discussed in introduction section to improve the background study. Recently, low velocity impact damage was investigated for continuous fibre composite sandwich core [Ref: Behaviour of continuous fibre composite sandwich core under low-velocity impact] and hybrid composite sandwich core [Ref: A novel hybridised composite sandwich core with Glass, Kevlar and Zylon fibres–Investigation under low-velocity impact]. Suggest to include them in introduction section with proper citations to improve the background study.

I would be happy to see the revised version to understand how these comments are being addressed.

Round 2

Reviewer 2 Report

I have no further comments